# In Vivo Evaluation of ECP Peptide Analogues for the Treatment of *Acinetobacter baumannii* Infection

**DOI:** 10.3390/biomedicines10020386

**Published:** 2022-02-05

**Authors:** Jiarui Li, Guillem Prats-Ejarque, Marc Torrent, David Andreu, Klaus Brandenburg, Pablo Fernández-Millán, Ester Boix

**Affiliations:** 1Department of Biochemistry and Molecular Biology, Faculty of Biosciences, Universitat Autònoma de Barcelona (UAB), Cerdanyola del Valles, 08193 Bellaterra, Spain; jiarui.li@e-campus.uab.cat (J.L.); Guillem.Prats.Ejarque@uab.cat (G.P.-E.); Marc.Torrent@uab.cat (M.T.); Pablo.Fernandez@uab.cat (P.F.-M.); 2Barcelona Biomedical Research Park, Department of Experimental and Health Sciences, Universitat Pompeu Fabra, 08002 Barcelona, Spain; david.andreu@upf.edu; 3Brandenburg Antiinfektiva GmbH, c/o Forschungszentrum Borstel, 23845 Sülfeld, Germany; kbranden@gmx.de

**Keywords:** ECP, AMPs, infection, murine model, Gram-negative bacteria, LPS

## Abstract

Antimicrobial peptides (AMPs) are alternative therapeutics to traditional antibiotics against bacterial resistance. Our previous work identified an antimicrobial region at the N-terminus of the eosinophil cationic protein (ECP). Following structure-based analysis, a 30mer peptide (ECPep-L) was designed that combines antimicrobial action against Gram-negative species with lipopolysaccharides (LPS) binding and endotoxin-neutralization activities. Next, analogues that contain non-natural amino acids were designed to increase serum stability. Here, two analogues were selected for in vivo assays: the all-D version (ECPep-D) and the Arg to Orn version that incorporates a D-amino acid at position 2 (ECPep-2D-Orn). The peptide analogues retained high LPS-binding and anti-endotoxin activities. The peptides efficacy was tested in a murine acute infection model of *Acinetobacter baumannii*. Results highlighted a survival rate above 70% following a 3-day supervision with a single administration of ECPep-D. Moreover, in both ECPep-D and ECPep-2D-Orn peptide-treated groups, clinical symptoms improved significantly and the tissue infection was reduced to equivalent levels to mice treated with colistin, used as a last resort in the clinics. Moreover, treatment drastically reduced serum levels of TNF-α inflammation marker within the first 8 h. The present results support ECP-derived peptides as alternative candidates for the treatment of acute infections caused by Gram-negative bacteria.

## 1. Introduction

It is now nearly one century since the landmark discovery of penicillin. Unfortunately, the emergence of antimicrobial resistance (AMR) is demanding the development of novel antibiotics [1]. Among hundreds of active molecules, we cannot disregard the potential of biomacromolecules such as antimicrobial peptides (AMPs), which are endowed with some unique properties distinct from small molecules, widely used in the drug development field [2,3,4].

AMPs are peptides that can be derived from organisms of all kingdoms and display a variety of functions. Bacteria produce AMPs to fight against other bacteria while animals can take advantage of AMPs to protect themselves. AMPs, and in particular cationic AMPs, the most populated group, have common properties and mechanisms because of their physicochemical nature. In particular, AMPs will exert their roles either by disrupting the negatively charged bacterial membrane or by targeting intracellular components within bacteria cells. Some native proteins or peptides are also able to regulate innate immunity, a feature that provides them a selective superiority over traditional antibiotics [5]. In addition, AMPs can contribute to the fight against multidrug-resistant (MDR) bacteria as well as biofilm communities [6,7,8]. Additionally, synergism between antibiotics and AMPs has been slowly identified [9,10]. Therefore, AMPs have a great potential for drug development. Recently, thanks to novel methodologies, many natural or synthetic AMPs have been proposed as antimicrobial drug candidates [11].

However, some drawbacks remain to be solved, such as their unpredictable toxicity or bio-stability, which can hinder their unconditional applicability. In particular, many discovered AMPs are only effective in vitro due to their lower stability in vivo. To overcome this issue, unnatural amino acid or non-coded amino acids have been introduced along the whole AMP sequence or at the sites sensitive to proteases [12,13,14,15]. Interestingly, AMP with L-amino acids replaced by D-amino acids can sometimes not only be highly resistant to proteolysis but also have better antibacterial activity than the L-version [16]. On the other hand, natural AMPs may have a too-long sequence, which will increase the cost of production. Thus, it is a good option to find active fragments from original AMPs and design shorter peptides that retain the antibacterial activity [17,18].

Despite some putative drawbacks that might remain to be addressed, many AMPs are already in clinical or preclinical trials. However, to date, most AMPs under clinical trial are only for topical application and the number of AMPs for systemic administration against drug-resistant bacteria are still a minority [18,19,20]. One representative AMP category is polymyxins, once discarded because of nephrotoxicity, but now reconsidered as the last resort for the treatment of infection caused by multi-resistant strains [21]. It is worth noting that polymyxins as non-ribosomal peptides contain non-standard amino acids and are, therefore, protected against proteolysis. In particular, polymyxin E (colistin), the most frequently used in the clinics, has a D-Leu in its sequence at position 6 [22]. It is important to highlight that colistin is produced in vivo by non-ribosomal peptide synthetase (NRPS) in which D-amino acids are incorporated through the action of the epimerization domain [23].

Our laboratory has been long-standing working on the structure-function of secretory ribonucleases (RNases) that belong to the RNase A superfamily, a vertebrate-specific family with 13 members identified in humans with diverse biological functions, including antimicrobial activity, immune modulation and host defense [9,24,25]. Among them, RNase3, also known as the eosinophil cationic protein (ECP), stands out as the family member with the highest antimicrobial activity, where an essential domain was identified at its N-terminal, with LPS-binding, cell-agglutinating and anti-biofilm properties [26].

Considering the outstanding performance of ECP native protein and derived peptides in vitro, we designed new ECP peptide analogues based on the 30mer lead template [27], that incorporate non-natural amino acids to limit proteolysis in human serum [28]. Next, two peptides were selected for further characterization and in vivo studies. The first is the all-D version of the ECPep-L lead peptide (ECPep-D) [27], where all L-amino acids in the sequence have been replaced by D-amino acids. The second peptide incorporates Orn substitutions for Arg residues and includes a D-amino acid at position 2 (ECPep-2D-Orn), based on the previous identification of ECP peptide main proteolysis target sites [28]. In addition, recent work confirmed that Arg to Orn replacement ensured the peptide stability in serum in vitro up to more than 8 h and retained its antimicrobial properties [28].

In the present work, we explored the efficacy of the two peptides analogues (ECPep-D and ECPep-2D-Orn) in a murine infection model. Based on our previous work, in vitro characterization against representative Gram-negative bacteria species, we selected *Acinetobacter baumannii* (*A. baumannii*) [29,30], which belongs to the ESKAPE top priority WHO pathogen list, as our working reference for the present in vivo murine infection model. To note, AMPs are currently regarded as one of the most promising candidates to treat *A. baumannii* infections.

## 2. Materials and Methods

### 2.1. Bacteria Strain, Cells, Mice and Other Materials

*A. baumannii* strain (CECT 452, Valencia, Spain, ATCC 15308, Manassas, VA, USA) and *Pseudomonas aeruginosa* (*P. aeruginosa*) strain (CECT 4122, Valencia, Spain, ATCC 15692, Manassas, VA, USA) were from the Colección Española de Cultivos Tipo (CECT, Valencia, Spain). The *Escherichia coli* (*E. coli*) BL21 (DE3) strain was from Novagen (Darmstadt, Germany). MRC-5 cells were from the American Type Culture Collection (ATCC, Manassas, VA, USA) and HEK293T cell line was kindly provided by Dr. Raquel Pequerul (UAB, Barcelona, Spain).

Balb/c mice (9–12 weeks old, 20–30 g, male and female, were supplied by Charles Rivers Laboratories and the study procedures have been approved by the Animal and Human Experimentation Ethics Committee at Universitat Autònoma de Barcelona (UAB, Barcelona, Spain). The animals were acclimated at least 5 days from arrival until the start of the experiment with free diet and water provided.

Lipopolysaccharides (LPS) from *E. coli* O111:B4 (O-LPS), EH100 (Ra mutant), and J5 (Rc mutant), porcine mucin, Mueller Hinton broth (MHB) and Thiazolyl Blue Tetrazolium Blue (MTT) were purchased at Sigma-Aldrich (Saint Louis, MO, USA). Colistin sulphate and Dimethyl sulfoxide (DMSO) were from Apollo Scientific (Stockport, UK). BODIPY^®^ TR cadaverine (BC) were from Molecular Probes (Eugene, OR, USA). Mouse TNF-α ELISA Kit (MBS175787, MyBioSource, San Diego, CA, USA) and Mouse LPS ELISA Kit (MBS7700668, MyBioSource, San Diego, CA, USA) were from bioNova científica s.l. (Madrid, Spain) Human TNF-α ELISA set was from BD Biosciences (555212, BD OptEIA^TM^, San Jose, CA, USA). Aspidasept^®^ (pep19-2.5) is an antibacterial/anti-inflammatory peptide drug (GCKKYRRFRWKFKGKFWFWG, with C-terminal amide) [31].

### 2.2. Peptides Synthesis

Peptides were synthesized at the Universitat Pompeu Fabra (UPF, Barcelona, Spain) Peptide Synthesis Service as previously described [27]. Briefly, peptides were assembled in C-terminal carboxamide form on an H-Rink Amide-Chem Matrix resin using Fmoc solid-phase peptide synthesis (SPPS) protocols. After chain assembly, peptides were fully deprotected and cleaved from the resin with TFA/H_2_O/Triisopropylsilane. Peptides were precipitated from the TFA solution by addition of chilled diethyl ether followed by three centrifugations at 4800 rpm, 5 min, 4 °C, taken up in water and lyophilized. Crude peptides were checked by analytical RP-HPLC and LC-MS and purified by preparative RP-HPLC as previously detailed [27]. Fractions of >95% HPLC purity and with the expected mass by LC-MS were pooled and lyophilized. Peptide purity was assessed by the area of the purified peptide peak relative to the total peak areas in the chromatogram. Peptide stock solutions were prepared in sterile deionized water and stored at −20 °C.

### 2.3. Circular Dichroism Assay

Far-UV CD spectra were obtained from a Jasco-715 (Jasco), as previously described [32]. The spectra were registered from 190 to 240 nm at room temperature. Data from four consecutive scans were averaged. Before reading, the sample was centrifuged at 10,000× *g* for 5 min. Peptide spectra were obtained at 16 µM in 5 mM Tris, pH 7.4 and 1 mM SDS, with a 0.2 cm path-length quartz cuvette.

### 2.4. Minimum Inhibitory Concentration (MIC) Determination

The MIC is defined as the lowest concentration of one reagent to prevent visible growth of bacteria. The MIC determination was followed by the protocol described previously [33]. Briefly, bacteria in exponential growth were used to prepare a suspension in MHB with the approximate number of 5 × 10^5^ CFU/mL. Next, an aliquot of 90 μL of bacteria suspension was added into each well of the 96-well polypropylene plate. Immediately, 10 μL of peptide diluted by 0.01% acetic acid was added to the corresponding well to have a final concentration ranging from 20 to 0.16 μM. The plate was incubated for 24 h at 37 °C, 100 rpm. The presence or absence of bacterial growth was visually inspected and confirmed by reading OD_600_ with Tecan Microplate Reader Spark^®^. Each test was performed in triplicate.

### 2.5. LPS Affinity Assay

The LPS affinity was assessed using the fluorescent probe BC by an adaptation of the displacement assay reported [27,34]. Peptides and colistin were serially diluted in a 96-well fluorescence plate from 20 to 0.16 μM in Tris/HCl 50 mM PH 7.4. Next, LPS and BC diluted in the same buffer were added to have the final concentration of 50 μg/mL and 5 μM, respectively. Fluorescence measurements were performed on Tecan Microplate Reader Spark^®^ at 580 nm of excitation wavelength and 620 nm of the emission wavelength at 5 nm for optimal gain. Each test was performed in triplicate.

### 2.6. Cytotoxicity Assay

Cytotoxicity was measured for the MRC-5 and HEK293T cell lines by MTT assay [35]. Cells were grown in 5% CO_2_ at 37 °C with MRC-5 cell line maintained in MEMα and HEK293T in DMEM/F-12 media, both of which were supplemented with 10% FBS. The cells were passaged in 25 cm^2^ or 75 cm^2^ flasks to prepare 96-well plates with 3 × 10^4^ cells/well, which were incubated overnight. Next, serial dilutions of ECPep-D and ECPep-2D-Orn were added to have a final concentration ranging from 100 to 0.78 μM (375 μg/mL to 3 μg/mL) and colistin was added at final concentrations ranging from 300 to 2.34 μM (380 μg/mL to 3 μg/mL) in corresponding wells. After incubation to the specified time, the medium of the plate was replaced by fresh medium containing 0.5 mg/mL MTT solution and the mixture was incubated for 2.5 h in 5% CO_2_ at 37 °C. The medium was then removed, and formazan was dissolved by adding DMSO. The optical density (OD) was recorded by using a Victor^3^ plate reader (PerkinElmer, Waltham, MA, USA) set at 540 nm and 620 nm as a reference. Each test was performed in triplicate.

### 2.7. Tolerance Study of Peptides in Mouse

Initially, the “Up and Down” protocol was followed to determine the maximum lethal dose with some modifications. Briefly, one mouse was taken per peptide and increasing doses of the peptide were injected through the intraperitoneal (*i.p.*) route every 48 h, and the concentration at which the mouse had severe clinical symptoms or died was determined (Figure 1a).

Once the lethal concentration had been determined, we continued with the “main study” (Figure 1b). Two groups of 6 female mice were used at two different concentrations of both peptides, one and two concentrations below the maximum considered either by lethality or by clinical signs. The toxicity was further verified by mimicking the doses and frequency foreseen in the previous study, i.e., two doses per day for 3 days as the initial plan. In all cases, the trials were stopped or readjusted owing to ethical principles when obvious suffering and sudden death of mice were observed in any of these groups.

There was also an extra group to confirm the non-toxicity of the vehicle. For extra caution, some additional doses were applied to confirm the safety dose in case of repeated injections (Figure 1c).

### 2.8. Mouse Systemic Infection Model

An aliquot from an *A. baumannii* stock stored in 15% glycerol at −80 °C was taken to culture in 3 mL LB overnight at 250 rpm, 37 °C. Next, OD_600_ reading of *A. baumannii* overnight culture was performed to adjust to the number of bacteria required (CFU/mL) and then diluted by half with the same volume of 10% porcine mucin, to obtain a solution of selected CFU/mL in 5% mucin for mice inoculation. Mucin was added to enhance the infectivity of *A. baumannii*, as previously established [36]. The inoculating volume was decided based on the body weight of each animal (10 mL/kg).

Following this, the evolution of the animals was supervised for 48 h by evaluating clinical signs and body weight as detailed in Section 2.10 (Appendix A). When the sum of the scores reached a maximum value of 7, the mice were euthanized following ethical principles. The infection was first tested at a concentration of 10^9^ CFU/kg and then set at 10^7^ CFU/kg and 10^8^ CFU/kg. Criteria for initial infection conditions were established as later detailed. We also performed an extra assay where mice infected by 10^8^ CFU/kg of *A. baumannii* were treated with colistin, used as a positive control.

### 2.9. Efficacy Assay of Peptides in Infection Mice Model

In the first assay, 24 mice (12 females and 12 males) were randomly distributed in 4 groups (3 males and 3 females in each group) with different treatments: vehicle (HBS buffer, negative control), ECPep-D (10 mg/kg), ECPep-2D-Orn (10 mg/kg) and colistin (15 mg/kg), the latter as a positive control. To avoid bias, the reagents for these four groups were given to technicians in advance and renumbered randomly, designated as A, B, C, D, respectively. The treatments corresponding to these codes were not to be told to the operators until the end of the experiment. The murine acute infection model followed the same protocol as introduced before with a lethal concentration (10^8^ CFU/kg) to generate sepsis. The treatment was applied 2 h later after the bacteria inoculum and once every 24 h for 3 days through *i.p.* injection. All the animals were checked for 3 days following the clinical signs described in Section 2.10 (Appendix A). On study day 3, surviving animals were euthanized by using an overdose of 200 mg/kg pentobarbital given intraperitoneally.

Following the first assay, we also performed a second assay, with the introduction of some modifications. Using the same infection model and injection route, another 46 mice were divided into 4 groups randomly, in which 14 for ECPep-D (20 mg/kg), 14 for ECPep-2D-Orn (20 mg/kg), 12 for colistin (15 mg/kg) and 6 for vehicle, with gender equally divided. The treatment was also applied 2 h later after the bacteria inoculum but only once during the total study. All the animals were also checked for 3 days as in the first assay.

### 2.10. Evaluation of Body Weight and Clinical Symptoms

Body weight (BW, g) of mice was measured before every injection and % BW gain was calculated.
% BW gain = (new BW − initial BW)/initial BW × 100%(1)

In the tolerance study, for the “Up and Down” assay, the BW was recorded twice on the administration day and, for the “main study”, it was measured every day at the first 3 days and then at some selected days until the 16th day, when the monitoring was finished. In both efficacy assays, BW was recorded once a day but on the first day (Day 0), a second measure was conducted at 8 h.

The clinical symptoms were assessed through a scoring system, which contains 9 items, including changes of body weight, behavior, breathing, skin, hair, eyes and gastrointestinal status (see full list at Appendix A). The score for each item increases from 0 to 3 according to the severe levels, up to a total of 27. During the study, when an animal obtained a score of 3 in any of the parameters assessed with a maximum score of 3, euthanasia was carried out to apply the endpoint criteria. Euthanasia would also be practiced if the sum of several parameters that are less than 3 separately was ≥ 7.

In the tolerance study, the clinical score was recorded at about 10 min, 30 min, 1 h, 2 h, 4 h, 7 h, 24 h and 48 h after injecting the peptides for the “Up and Down” assay and in the main study, the clinical score were only monitored when the BW was measured. During the efficacy assay, the clinical score was recorded every 2–3 h on Day 0 and then twice every day.

### 2.11. Assay of CFU in Mice Tissues

The evaluation of CFUs in extracted tissues, spleen and lung, was performed at the end of the treatment. In both efficacy studies, one lung and half spleen were collected at the time when each animal was executed due to a high clinical score or euthanized at the end of the study. The organs were homogenized in HBS with 6 successive 1:10 dilution after being weighed on a precision scale. Each dilution was seeded in Petri dishes with LB-agar and the colonies were counted after 16 h of incubation at 37 °C. The final CFUs for each organ were calculated with respect to the weight of organs (CFU/g).

### 2.12. Quantification of TNF-α and LPS in Mice Serum by ELISA

In the second efficacy study, the blood of animals that either died for severe clinical symptoms or were euthanized at the end of the assay, were collected for testing TNF-α and LPS levels by ELISA. In order to check the levels of TNF-α and free LPS, only serum was analyzed, after discarding blood cells by centrifugation. The blood from mice was centrifuged at 3200 rpm for 30 min and the supernatants were collected carefully to ensure that there were no sediments left. Mouse ELISA kits for TNF-α and LPS were used.

According to the description of the manufacturer, LPS level determination is based on the principle of double antibody sandwich technology, where the microplates pre-coated with specific antibodies were used. Following addition of serum samples to the wells, the HRP-Conjugate reagent was added to form an immune complex. After incubation and washing, the unbound enzyme was removed. Then, chromogen substrates were added successively and optical density was recorded using microplate reader set at 450 nm. Sample readings were corrected using the baseline corresponding to serum of mice neither infected nor treated. A calibration standard curve was prepared and final calculated values were positively correlated with the concentration of LPS. Two replicates were performed according to the protocols for the representation of the standard curve and the sample testing.

### 2.13. Stimulation of Human MNC by LPS and Endotoxin Neutralization Assay

The stimulation assay of LPS R60 HL185 on the Mononuclear cells (MNC) for the TNF-α response was based on the previously used method [31]. Mononuclear cells were isolated from heparinized blood samples obtained from healthy donors as described [37]. The cells were resuspended in 1640 RPMI medium and their number was equilibrated at 5 × 10^6^ cells/mL. For stimulation, a total of 200 μL (1 × 10^6^) of MNC was plated in each well of a 96-well plate and the cells were then stimulated by LPS Ra (from *S. Minnesota* strain R60) alone or at selected peptide/LPS ratio, which were pre-mixed during 30 min at 37 °C. After 4 h of incubation at 37 °C with 5% CO_2_, samples were centrifuged to collect the supernatants to determine the level of TNF-α by ELISA kit (BD OptEIA^TM^). Two replicates were performed in each test.

### 2.14. Statistical Analysis

A Mann—Whitney test or unpaired t-test was used for statistical comparison depending on whether the sample size between groups was equal. EC_50_ or IC_50_ was calculated by nonlinear fit with normalized response. The ELISA standard curve was fitted by linear regression (Appendix A).

## 3. Results

### 3.1. Design and Structural Characterization of ECP Peptide Analogues

Two peptide analogues, named ECPep-D and ECPep-2D-Orn (Figure 2), were selected based on the ECP reference 30mer peptide (ECPep-L). ECPep-D was designed and optimized based on the identification of the main structural determinants for antimicrobial activity against Gram-negative planktonic and biofilm cultures, as previously detailed [26,27]. The 30mer sequence includes an aggregation-prone region (A6-I14) that promotes bacterial agglutination and membrane lysis and a region involved in the LPS binding (Y27-R30) [27]. In addition, the reference peptide is highly cationic and includes 6 Arg (Figure 2). Based on a target proteolysis study in serum, peptide analogues were synthesized with either all D-AA, Arg to Orn substitutions or blockage of peptide bond cleavage by D-AA introduction. Additionally, based on the previous analysis of the proteolysis digestion products [28], ECPep-2D-Orn was selected. ECPep-2D-Orn is the Orn-peptide analogue, which showed the highest half-life in serum and incorporates a D-AA at position 2 (Figure 2). The RP-HPLC and MS chromatograms of these peptides are shown in Appendix A.

Structural analysis in aqueous solution and in the presence of lipid dodecyl phosphocholine (DPC) micelles was previously assessed by NMR [28]. NMR indicated that the 30mer L-reference peptide adopts in aqueous solution a defined helix from residues 5 to 13 and tends to helical structuration within the 16–27 stretch. In the presence of DPC the peptide gets more structured. A similar pattern was observed for ECPep-2D-Orn by NMR analysis [28]. In addition, a comparison of all-L and all-D peptides by CD confirmed that equivalent secondary structures are adopted by both enantiomers (Figure 3).

### 3.2. In Vitro Activities of ECP Peptides

First, the antibacterial activity was tested in vitro. Overall, there were no major differences among the calculated MIC for the three Gram-negative species tested, with values of 10 μM, except for ECPep-2D-Orn that displays a significant enhanced effectivity against *A. baumannii* (Table 1). All data were compared to colistin peptide, used as a positive control. In addition, the survival percentages of each species exposed to the peptides at sublethal doses were compared (Figure 4). Comparison of the bacteria survival percentage upon exposure to a range of peptide concentrations at ≤MIC_100_ showed considerable reduction in the bacterial viability and highlighted significant differences between the peptides’ relative activities. Overall, the results indicated that when the bacteria were exposed to the peptides at different concentrations below the MIC value, the growth was altered in a dose-dependent mode. In particular, the results highlighted the best performance for ECPep-D, which can significantly inhibit the growth of *E. coli* and *A. baumannii* between 0.625 and 1.25 μM, in comparison with the original L-version (ECPep-L) and Orn analogue (ECPep-2D-Orn) that required concentrations above 2–5 μM.

Following this, the relative LPS affinity of ECP peptides and colistin were compared. Comparison of relative binding affinities of the peptides with three different types of LPS extracted from *E. coli*, indicated that the ECP peptides have a lower or similar EC_50_ in comparison to colistin. Overall, ECP peptides showed better performance than colistin at the micromolar concentration range, with a higher affinity for the full O-LPS structure, followed by the LPSRa type, which lacks the O-antigen. On the other hand, both L- and D- peptide versions showed lower binding ability on LPSRc, the type with the shortest structure among the three tested LPS; where only the Orn-version was slightly better than colistin for the truncated LPS type (Table 2, Appendix A). Interestingly, no significant differences were appreciated between the calculated EC_50_ values for all-L and -D peptide versions, indicating that the all-D peptide could adopt an equivalent overall conformation essential for LPS binding.

Next, we tested the cytotoxicity of the peptides. For the fibroblast MRC-5 cell line, all peptides hardly showed any toxicity after 4 h of incubation. Moreover, no toxicity was observed for ECPep-2D-Orn at the highest tested concentration during all the assayed time range (up to 48 h). However, for the ECPep-D, we estimated an LD_50_ below 50 μM at 24 or 48 h exposure time (Table 3, Appendix A). In contrast, the LD_50_ of ECPep-D for kidney HEK293T cell line at 48 h was significantly higher (LD_50_ ≈ 63.60μM) than that for MRC-5 (LD_50_ ≈ 23.52 μM) (Appendix A). Overall, the values for ECP peptides are similar or slightly lower than colistin, at short and long exposure time, respectively.

To sum up, we can conclude from the in vitro studies that the activities of the two analogues (ECPep-D and ECPep-2D-Orn) are similar or even better than the original version peptide (ECPep-L), indicating that these two analogues are valuable candidates for in vivo assays. In addition, previous studies demonstrated that the half-life time of ECPep-L in human serum is significantly shorter in comparison to that of ECPep-2D-Orn (12 min vs. > 480 min) [28]. Therefore, we decided to use ECPep-D and ECPep-2D-Orn to conduct the follow-up in vivo research.

### 3.3. Tolerance of Peptides in Mice

Next, we proceeded to evaluate the peptides potential toxicity in vivo using a mice model. Firstly, an “Up and Down” protocol was carried out to obtain an initial estimate of the maximum tolerated dose for both peptide analogues following *i.p.* injection. We started the tolerance assay for each peptide at a low dose and increased it slightly every 48 h provided that the animal did not show any obvious suffering. Both mice had a clinical score over 6 within 30 min when the dose of peptides reached 30 mg/kg, estimating the maximum tolerated dose for ECPep-D at 20 mg/kg and for ECPep-2D-Orn at 25 mg/kg with one single dose from the “Up and Down” assay (Appendix A). These results were later confirmed by the “main study” in which a single dose of 20 mg/kg for ECPep-D and 25 mg/kg for ECPep-2D-Orn did not cause any obvious clinical signs in all mice.

However, severe clinical suffering was observed when the dose was applied repeatedly in mice at concentrations from 15 mg/kg to 25 mg/kg (Appendix A). Therefore, the initial dosing plans were modified or suspended accordingly (Appendix A). On the other hand, no extra death occurred and the clinical symptoms were recovered for the survival mice after the treatment suspension. The mice’s healthy condition lasted till the end of the observation period (16 days).

In addition, within the extra group with low dose (7.5 or 10 mg/kg) administration, no obvious clinical symptoms were observed (Appendix A).

### 3.4. Murine Acute Infection Model by A. baumannii

Once the toxicity assay was finalized, we proceeded to evaluate the efficacy of the peptides in an *A. baumannii* infection model. All the mice showed very severe suffering after inoculation with 10^9^ CFU/kg of *A. baumannii*, corresponding to 10^8^ CFU/mL of the initial concentration of bacteria supplemented with 5% porcine mucin. Therefore, euthanasia had to be applied according to the protocol endpoint criteria after 7 h. Following this, a lower level of infection was applied at 10^7^ CFU/kg in a small test group (1 male and 1 female). In this assay, both mice only experienced slight suffering during the first few hours and totally recovered on the next day. Therefore, an in-between concentration of 10^8^ CFU/kg was selected for further analysis, where clinical suffering was lower than the one observed at 10^9^ CFU/kg (Appendix A). Moreover, the group of infected mice at 10^8^ CFU/kg was treated with the positive control (colistin at 15 mg/kg). A slight increase in clinical signs appeared after the first 8 h and it was nearly undetectable after 24 h in the surviving mice (Appendix A).

Thus, we decided to use 10^8^ CFU/kg as the mice inoculation concentration of *A. baumannii* in the following assays and established these conditions for the infection model to test the efficacy of the peptide.

### 3.5. Treatment with Peptides Improves the Survival Rate and Relieves Clinical Suffering of Infected Mice

To assess the therapeutic effect of our peptides on infected mice, we performed two assays with different survival rates in peptide treatment.

In the First Efficacy Assay, we applied a peptide/mouse of 10 mg/kg once per day, a previously established safe dose, and the treatment lasted for 3 days. In the vehicle-treated group (negative control), as already observed in the set-up experiment (Appendix A), all animals presented severe clinical signs within the first 8 h post-inoculum (Appendix A) and were, therefore, euthanized after the first day. On the other hand, all the animals treated with colistin (positive control) had a 100% survival rate (Figure 5a). In this group, although a decrease in weight was observed during the first day following infection, the mice recovered weight during the assay time course, which was considered a sign of health improvement. Moreover, we also observed only a slight increase in the clinical signs score, indicative of health problems during the first 24 h, being subsequently minor or non-significant during the following observation period (up to 3 days) (Appendix A). Surviving animals treated with the ECP peptides exhibited the same behavior pattern as the colistin-treated animals (Appendix A). Although survival rates for mice treated with 10 mg/kg of the ECP peptides ranged 70–80% at the first 8 h of observation, the values drastically dropped when checked at 24 h. Overall, the survival rate after the three-day treatment was equivalent for both all D and 2D-Orn peptides, but much lower than the colistin positive control (Figure 5a). However, the weight gain of both mice was equal to the curve obtained with the colistin group. The score of the clinical signs of the surviving animals treated with ECP peptides also had similar values to the positive control group (colistin), signs of clinical recovery together with weight gain (Appendix A). In any case, no direct side-by-side comparison could be performed here between colistin and ECP peptides efficacy, as the assay concentration was not equal: 10 mg/kg for the ECP peptides versus 15 mg/kg for colistin, the latter selected to ensure a 100 % survival positive control.

Subsequently, in a second efficacy assay, we decided to test a higher peptide concentration (20 mg/kg), a value where no toxicity was observed for a single dose in the previous toxicity assay (Appendix A). Additionally, the positive control (colistin) was kept at the same dose as before but with only a single injection to follow the same one-time injection protocol selected for the ECP peptides’ treatment. Following an equivalent pattern as previously observed for the vehicle group, all animals without any treatment presented severe clinical signs at the first 8 h / post inoculum and were, therefore, euthanized. On the other hand, the 12 animals treated with colistin had a 100% survival rate (Figure 5b), as previously registered. When comparing the evolution of the clinical sign scores we observed first an increase during the first 8 h due to the process of infection (Figure 6). The decrease in body weight was also observed on the first night and then mice started regaining weight during the rest of the experiment time course. Interestingly, we observed a significant improvement of clinical scores and recovery of body weight in the mice groups treated with both peptides in comparison with the negative control, treated only with Vehicle (HBS) (Figure 6). Meanwhile, the score of the clinical signs of the surviving animals treated with the peptides had similar values to colistin: both signs of clinical recovery and body weight gain.

However, the survival curve of each ECP peptide treatments revealed a differentiated efficacy profile (Figure 5b). A significant improvement for the treatment with ECPep-D at 20 mg/kg was achieved with a survival rate of 71% at 8 h and this survival rate lasted till the end of the study (72 h). On its side, the survival rate after 3 days for ECPep-2D-Orn was only 14%, which correspond to a similar value to previous results obtained at 10 mg/kg for both peptides.

### 3.6. ECP Peptides Reduce Bacterial Counts in Mice Organs and TNF-α Levels after 8 h of Treatment

In both efficacy assays in *A. baumannii*-infected mice, all the animals from the vehicle-treated group (negative control) were killed after 8 h due to the bad clinical symptoms, as evaluated by the calculated clinical score (Appendix A). Following this, the lung and spleen were collected and the infection level was analyzed by CFU counting. The number of calculated CFUs showed maximum values of 10^9^–10^10^ CFU/g. On the contrary, in positive controls where the mice survived 3 days, colonies counted after euthanasia at the end of the study presented a minimum value in all groups, around 10^3^ CFU/g (Appendix A and Figure 7a). Complementarily, blood samples of each sacrificed animal in the second efficacy assay were also collected and levels of TNF-α and LPS in serum were evaluated. For comparative purposes, blood samples corresponding to the same timing were analyzed together: either Day 0 (8 h) or Day 3 (72 h). Due to the differences in the number of animals that could be tested in each treatment group, the Mann—Whitney test was applied.

Results indicated that even if the efficacy to increase the survival rate between the two peptides differed greatly (about 70% for ECPep-D versus 20% for ECPep-2D-Orn at the first 8 h), the ability to reduce CFU counts in the studied tissues and the TNF-α inflammation marker levels in serum were similar for both peptides at the end of the assay (3 days). Therefore, we can conclude that both peptides were able to drastically reduce the CFU counts and TNF-α serum levels equivalently to colistin, the positive control. To note, only a slight significant reduction in organ CFUs was observed at Day 0 (*p* < 0.05) but CFUs counts at Day 3 were equivalent for both peptides and colistin (Figure 7a). Interestingly, both ECP peptides were able to lower serum TNF-α values at day 0 (8 h) more than twice with respect to the negative control (Figure 7b). Overall, when the mice survived at the end of the assay (Day 3) in both peptide-treatment groups, the number of colonies and the concentration of TNF-α were equivalent to the positive control.

On the other hand, regarding the LPS quantification, a different profile was observed (Figure 7c). On Day 0, the concentration of LPS in serum of sacrificed mice in all groups was undetectable. However, on Day 3, we observed an increase in the amount of LPS in both peptide-treatment groups. Moreover, the concentration of LPS was slightly increased in the colistin-treatment group respect to ECPep-D (*p* < 0.1), even though the mice seemed totally recovered with no clinical symptoms.

### 3.7. ECP Peptides Reduce TNF-α Levels in LPS-Stimulated Cells

Finally, we decided to evaluate the anti-endotoxin activity of the assayed peptides to complement and better interpret our in vivo results. To this end, we decided to compare the activity of ECPep-D (the peptide that showed better efficacy in vivo) with the original L-version (ECPep-L) and the colistin reference control. To evaluate the anti-endotoxin activity, we estimated the TNF-α levels in LPS-stimulated mononuclear cells. The results showed that both L- and D-peptide versions can significantly reduce the TNF-α levels when cells are exposed to LPS stimulation (Figure 8). The assay was performed at two peptide/LPS ratios and cell stimulation by three LPS concentrations. Overall, the ECP peptides’ efficacy was lower than the positive control, colistin, although this difference was mostly observed when MNC cells were stimulated with a higher LPS concentration (10 ng/mL). Complementarily, we performed an additional comparative assay using an anti-endotoxin peptide (Aspidasept^®^), a promising peptide candidate to prevent septic shock [38]. Interestingly, at high concentrations of LPS (10 ng/mL), ECP-derived peptides present slightly better anti-inflammatory activity than Aspidasept^®^, which in turn shows a better performance at the lower LPS concentration (1 ng/mL) (Appendix A). Overall, all peptides (ECPep, Aspidasept^®^ and colistin) have similar activity at the lowest LPS concentration used (Figure 8 and Appendix A). These results back up the observed ability of ECP-derived peptides to reduce TNF-α levels in the studied mice infection model. Finally, an additional assay was performed at even higher concentrations of LPS (100 ng/mL), which induced the triple of TNF-α release (≈2300 pg/mL) in cells without peptide treatment (Appendix A). Results highlighted that both ECPep-L and ECPep-D at 10:1 ratio with respect to LPS significantly reduced the release of TNF-α, at a similar percentage to Aspidasept^®^, although the latter was more effective at a 100:1 ratio.

## 4. Discussion

To achieve a proper assessment of the potentiality of novel antimicrobials as drug candidates, in vivo studies are mandatory. Following promising results in vitro, animal models are essential for monitoring both the bacterial infection process and the host response. Mice infection models can provide valuable information about how diseases or treatments may behave in humans, thanks to the similarities in the mammalian tissue structures and the functioning of their immune system. In particular, in vivo assays are essential for the evaluation of AMP agents. Firstly, many AMPs that achieve good MIC values in vitro may show bad or no effect in vivo due to their rapid degradation by proteases in the body. Secondly, the observed toxicity in vitro in cell line assays may become weak in animals thanks to the body’s self-regulation. In addition, the immune response in the body can potentiate the mechanism of action of AMPs and overcome bacterial resistance systems. Moreover, by testing the AMP efficacy in animal models, we will gain an understanding of their functional role as key players of the innate host defense response [39]. Among animal studies, murine bacterial infection models are probably the best characterized for evaluation of AMP activity [40,41,42].

Our laboratory has long been committed to the research and development of novel antimicrobial peptides based on the structure-functional knowledge on RNase A superfamily. Extensive studies of previous work have confirmed that RNase3 (ECP) displays the highest antibacterial activity among human family members [43,44]. Based on structural analysis, proteolysis mapping and peptide synthesis we identified at the N-terminus a region that retained most of the parental protein antimicrobial activity. Following a minimization effort and the identification of the sequence determinants for bacteria cell wall binding, membrane lysis, protein aggregation and cell agglutination, a 30mer was selected as the best pharmacophore with high antibacterial activity in vitro against both planktonic and biofilm cultures of Gram-negative bacteria [26,27]. Next, ECP peptide analogues were designed to ensure protection against potential in vivo proteolysis [28]. The incorporation of non-natural amino acids has been previously reported as a successful strategy to enhance AMPs biostability while retaining their antimicrobial properties [45,46]. In the current study, aiming to develop a potential AMP drug candidate, we selected two N-terminal derivatives of ECP (5–17P24–36), intended to enhance its biological stability in vivo. Based on the original L-version of peptide ECP (5–17P24–36)], named ECPep-L, we synthesized an all D-version peptide (named ECPep-D) together with the best analogue from recent biostability assays [28], where all Arginines were substituted by Ornithine and peptide bond at position 2 was protected from proteolysis by D-amino acid substitution, named ECPep-2D-Orn (Figure 2). Our previous work indicated that shielding the peptide bond between Pro2 and Phe3 by D-replacement significantly enhanced the peptide half-life in vitro, while retaining the antimicrobial properties of the parental ECPep-L and all Orn L-version (ECPep-Orn) [28]. Here, we incorporated the characterization of the peptide all-D version, showing equivalent MIC values in comparison to the all-L version for the three tested Gram-negative species (Table 1) but potential cytotoxicity in vitro at long exposure times (Table 3). It is worth emphasizing that while the D-version has an antibacterial activity equivalent to the L-version against the three selected Gram-negative species, the 2D-Orn-version was mostly effective against *A. baumanii* but less successful against the other two bacterial species. In any case, bacterial survival curves at 24 h using sublethal doses indicated that both peptide analogues (ECPep-D and ECPep-2D-Orn) had a better performance in *A. baumannii* culture, with the highest activity for the all-D version (Figure 4). Unfortunately, as commented above, ECPep-D shows significant cytotoxicity in contrast to ECPep-2D-Orn in the tested human cell lines at 24 h and 48 h (Table 3). Therefore, we decided to compare both peptide analogues (showing either high antimicrobial activity or no toxicity at the highest tested concentration 100 µM) in an *A. baumannii* mice infection model.

Interestingly, the efficacy of the peptides in a murine acute infection model by *A. baumannii* was encouraging. More than 70% survival after 3 days of infection was achieved by a single dose of the D-peptide analogue. In addition, in vivo toxicity test indicated that only significant clinical symptoms appeared when the peptide was administered at very high concentrations, such as 25–30 mg/kg, after repeated dosing. The maximum tolerated dose and the symptomatology after repeated doses observed in our work are similar to other reported cases for AMPs administered by the intraperitoneal route. Taking colistin sulfate as a reference approved AMP, we found a reported LD_50_ by intraperitoneal administration around 20–30 mg/kg in Swiss albino mice following a single injection [47]. Additionally, at the U.S. National Library of Medicine—Toxicity information (https://chem.nlm.nih.gov/chemidplus/rn/1264-72-8, accessed on 1 December 2021), the LD_50_ of colistin sulfate for the intraperitoneal route is 21.8 mg/kg in mice. It has been reported that mice receiving 32 mg/(kg × day) became sluggish after three doses of colistin, where three of twelve mice died from neurotoxicity [48]. Thus, we can conclude that in our studies with one single dose below the maximum tolerated dose, both ECP peptide analogues are safe for mice. Nevertheless, it is worth considering further work on novel Orn-analogues to enhance their in vivo biostability while minimizing their potential toxicity.

Next, we tested ECP peptides in an acute infection model. According to the literature to achieve an acute infection of *A. baumannii* in a murine model, the recommended dose of the inoculum ranges between 10^6^ CFU/mL and 10^8^ CFU/mL [49,50,51,52]. Following optimization by testing an inoculum by intraperitoneal administration from 10^6^ to 10^8^ CFU/mL, we selected a 10^7^ CFU/mL condition, where untreated mice developed severe clinical signs as well as heavy infection and had to be euthanized within a period of up to 8 to 9 h. On the other hand, a positive control was set by the administration of colistin at 15 mg/kg, where a drastic reduction in infection and recovery of clinical signs was achieved, reaching 100% mice survival. In addition, a supplement of 5% mucin was administered to boost infection and promote a proinflammatory response [36].

Subsequently, we started to test the potential efficacy of our designed peptides for acute systemic infection. Single doses from 10 mg/kg to 20 mg/kg, previously determined as safe, with no associated toxicity, were administered intraperitoneally to infected mice. A pattern of significant improvement of clinical parameters and recovery of body weight following ECP peptide administration was similar to the positive colistin-treated group. In contrast, in the negative non-treated control, none of the animals survived. The first efficacy assay at 10 mg/kg was followed by a second study at 20 mg/kg, where, in addition, a higher number of treated animals was inspected. A therapeutic effect of ECPep-D was observed with a 71.43% survival rate of mice after 3 days. In addition, a drastic reduction in CFUs was observed in the studied organs in all animals treated with both peptides, indicating that ECP peptides could be considered as potentially effective candidates for *A. baumannii* acute infection. However, although CFU levels at analyzed organs and clinical parameters recovery were similar to our positive control, a significantly lower survival ratio was reached for ECPep-2D-Orn peptide. This might be attributed to a poorer biostability in in vivo conditions of this analogue in comparison to the all-D peptide. Further work is envisaged to identify the best pharmacophore that ensures non-toxicity in vivo while retaining antimicrobial action. Complementary conjugation of ECP peptides to nanocarrier systems would be considered to reduce the effective dose and facilitate the targeted delivery, as demonstrated effective for a modified version of the parental protein [9].

Another interesting feature characteristic of ECP and its N-terminus derived peptides relies on its high binding affinity to lipopolysaccharides (LPS) present at the Gram-negative outer membrane [26]. Previous structural and functional studies characterized the protein and peptide interaction with LPS in vitro by the use of complementary approaches. The main protein residues that participate in the binding to LPS were characterized by side-directed mutagenesis, peptide-array library and NMR structural analysis [27]. The ECP targeted sequence for endotoxin binding is located at the C-terminus of the 30mer ECPep (i.e., YRWR) (see Figure 2). The present data highlights that the all-D peptide retains the same LPS binding affinity as its L counterpart (Table 2), suggesting that the overall structural determinants of the parental peptide are retained. More importantly, both ECP peptide analogues (ECPep-D and ECPep-2D-Orn) can drastically reduce TNF-α levels in mice serum and demonstrates for the first time the ability of an ECP peptide to block in vivo the release of the pro-inflammatory TNF cytokine. Nonetheless, our preliminary results on LPS determination in mice serum are less straightforward to interpret and would probably need future complementary assays. Despite the intrinsic limitations of the assay, where only a reduced number of samples could be assayed and unspecific interactions of the serum components with the ELISA kit reagents cannot be discarded, some results could be drawn when comparing the distinct analyzed groups. At Day 0 (8 h), nearly no LPS was detected within the serum of infected mice including the non-treatment group. This may be attributed to the fact that the unreleased LPS might remain on the surface of bacterial cells and would precipitate when the collected blood was centrifuged. Interesting results come out on Day 3. The amount of free LPS in the serum of mice cured with colistin is slightly higher on Day 3, which means that the LPS released from the bacteria remained in the serum and could not be neutralized after 3 days of infection. It has been reported that a large amount of LPS released from *P. aeruginosa* dead cells following treatment with colistin were still detectable, which in turn might reduce the effectiveness of this drug at the infection focus [53]. Indeed, LPS aggregates were reported as the main entities biologically active in previous studies [54]. In contrast to colistin, our designed peptides seem to be able to lower the blood circulating LPS after 3 days of treatment (Figure 7), although this issue needs further exploring. Among others, we might consider exploring the peptide anti-endotoxin activity in another animal model, such as rabbit, considered more appropriate for the study of sepsis-related symptoms [38]. In addition, a full understanding of the LPS-neutralizing process needs to take into consideration which is the biologically active conformation of blood circulating LPS [54] and how the AMP presence can alter the pro-inflammatory cell response.

Massive production of pro-inflammatory cytokines, such as TNF-α, are associated with the septic shock, a major lethal factor in Gram-negative infections in the clinics. Currently, colistin is frequently used as a last resort to treat high-risk patients that can undergo septic shock following acute infection. However, colistin is a non-ribosomal cyclic-AMP only used as a last resort due to its associated toxicity following chronic treatment [22,48]. Therefore, a new generation of antibacterial drugs that not only kill the bacteria but also neutralize LPS with no toxicity in vivo is essential. The present results corroborate the efficacy of ECP-derived peptides in vivo, showing their ability to effectively lower the number of CFUs in tested organs (lung and spleen) and the amount of TNF-α in the serum of infected mice (Figure 7). Moreover, both peptides administration can revert the severe clinical parameters during the first 8 h of infection and alleviate the mice suffering on the first day (Figure 6). In addition, the observed endotoxin neutralization activity of ECP peptides in vivo was corroborated in a human mononuclear cell assay, showing comparable values to Aspidasept^®^ (pep19-2.5), an anti-septic AMP drug candidate (Figure 8 and Appendix A).

## 5. Conclusions

The current results highlight the efficacy of two N-terminal derived peptides of ECP in a murine systemic *A. baumannii* acute infection model. This is the first report of an efficacy test of an ECP peptide in an animal model. Our data indicates that the 30mer all-D peptide version (ECPep-D) successfully enhances the mice survival rate up to 71% after 3 days. In addition, both peptides can recover the body weight and supervised clinical parameters to almost normal values, following an equivalent pattern as shown for animals treated with colistin, our assay positive control. Moreover, both ECP peptides reduce the release of the pro-inflammatory cytokine TNF-α. The present in vivo results make us confident to further explore the unique advantages of ECP-derived peptides as a new generation of antimicrobial candidates.

## Figures and Tables

**Figure 1 biomedicines-10-00386-f001:**
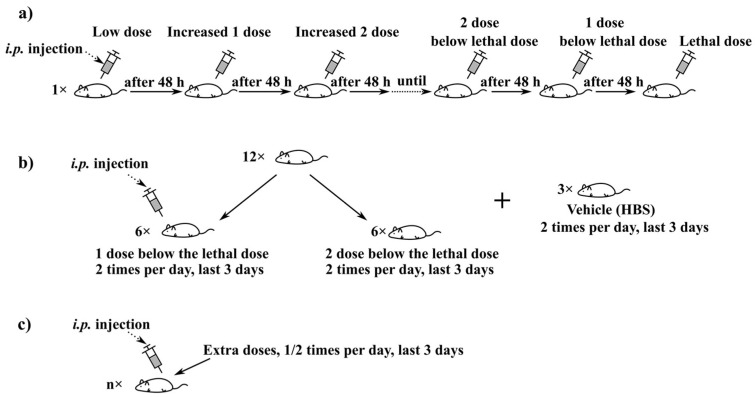
Toxicity study plans. (**a**) “Up and Down” protocol: 1 mouse was injected, starting at a very low peptide dose and observed at 48 h intervals. If the mouse showed no affected signs, with the same mouse, the dose was slowly increased every 48 h until it reached the lethal dose for this mouse. (**b**) Main study: mice were divided into groups randomly; 1 as well as 2 peptide doses below the lethal dose from the “Up and Down” assay was given two times per day for 3 days (the protocol was readjusted in case of emergency owing to ethical principles, as detailed in the results section and Appendix A). An extra group is for the vehicle. (**c**) Last study: Peptide was tested in smaller groups at lower doses depending on the results obtained from the main study.

**Figure 2 biomedicines-10-00386-f002:**
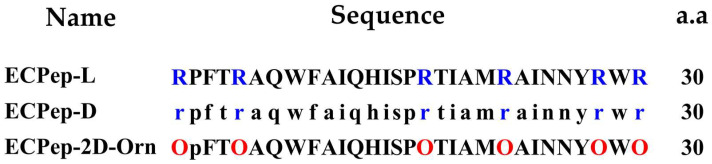
Sequence of the peptides. The amino acid sequence is indicated. L residues are indicated with an uppercase letter and D residues with lower case letters. Arginine or ornithine are colored in blue or red, respectively.

**Figure 3 biomedicines-10-00386-f003:**
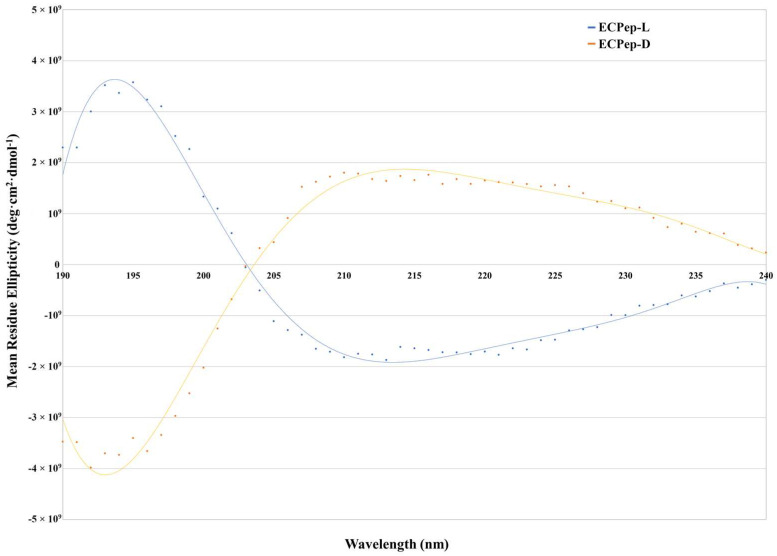
Circular dichroism spectra of ECPep-L and its D-enantiomer. Measures were performed in Tris 5 mM pH 7.4, 1 mM SDS and a peptide concentration of 16 µM using a Jasco J-715 spectropolarimeter.

**Figure 4 biomedicines-10-00386-f004:**
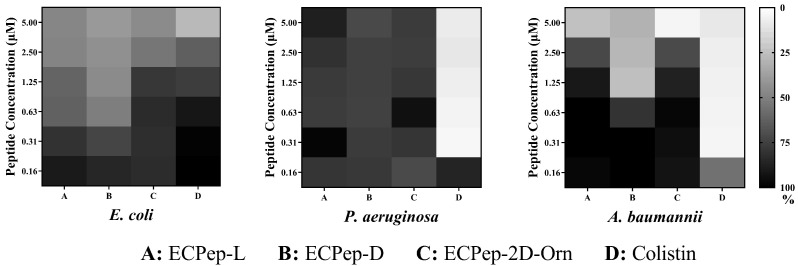
Heat maps of survival percentage of bacteria after 24 h incubation exposed to sublethal doses. The average survival percentages of bacteria normalized by the bacteria-free control and no-treated control are shown. The OD_600_ value equal to that of bacteria-free control is assigned as 0% and that of non-treated control as 100%.

**Figure 5 biomedicines-10-00386-f005:**
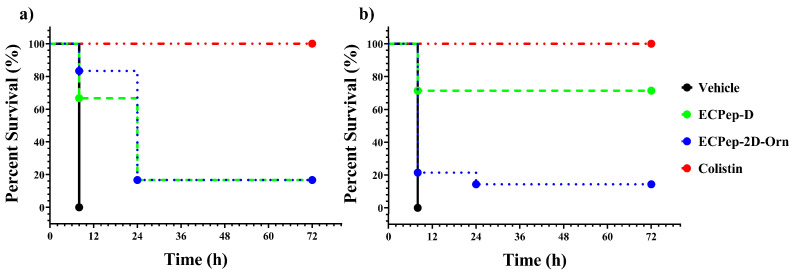
Survival curve of first and second efficacy assays. (**a**) Survival curve of infected mice treated with 10 mg/kg peptides. (**b**) Survival curve of infected mice treated with 20 mg/kg peptides. In both groups, each animal had been inoculated with 10^8^ CFU/kg *A. baumannii* with 5% mucin before treatment and the administration of colistin set at 15 mg/kg. The peptides were administered 2 h after the bacteria inoculation. The survival mice had been monitored for 72 h and euthanized.

**Figure 6 biomedicines-10-00386-f006:**
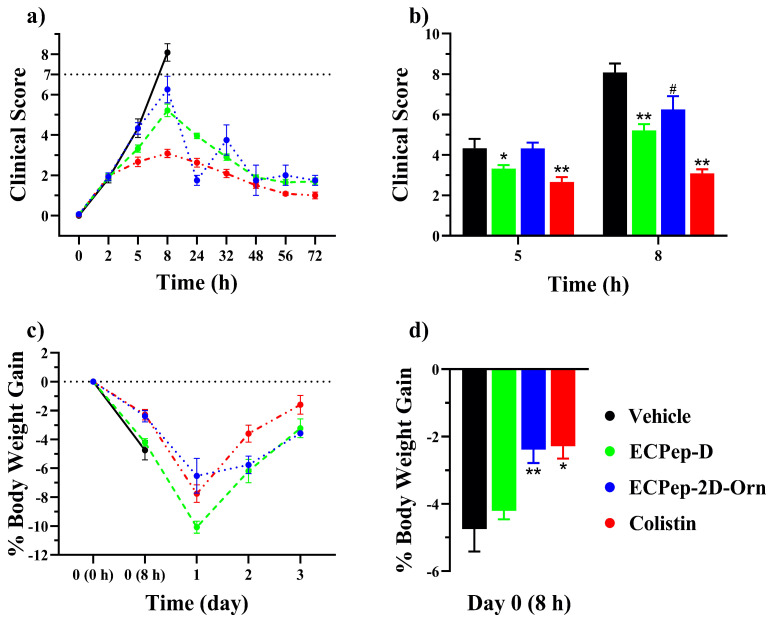
Changes of clinical symptoms during the 2nd efficacy assay with 20 mg/kg peptides treatment. (**a**) The average clinical scores in 3 days; (**b**) Histogram of clinical scores at 5 h and 8 h; (**c**) The time course body weight gain (%) up to 3 days; (**d**) Histogram of body weight gain (%) at 8 h. Each animal had been inoculated 10^8^ CFU/kg *A. baumannii* with 5% mucin before treatment. In all cases, a single dose of the peptide was given 2 h later than the bacteria inoculation. Mann–Whitney test has been used for statistical comparison between different treatments and vehicle (*** p* < 0.01, ** p* < 0.05, *^#^ p* < 0.1).

**Figure 7 biomedicines-10-00386-f007:**
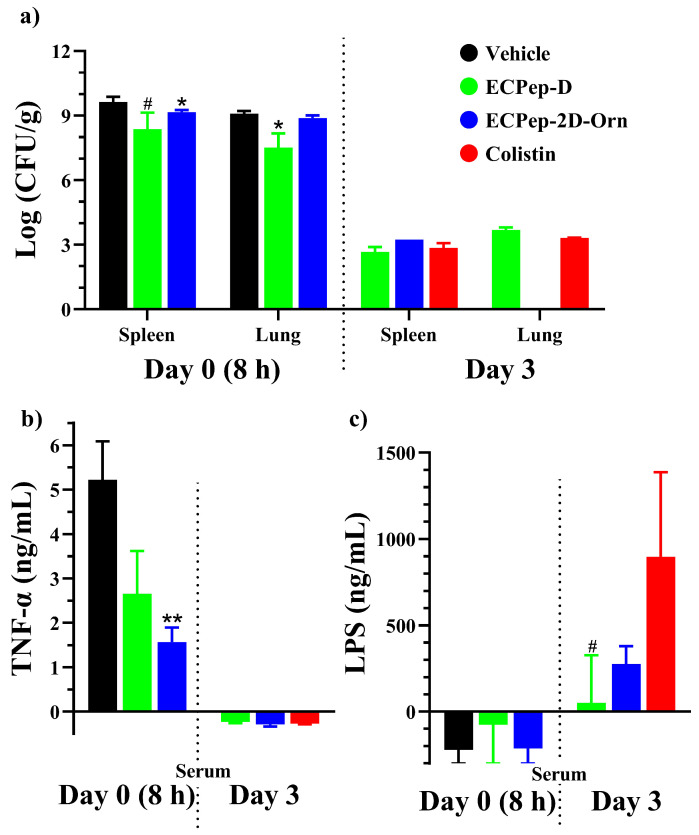
Evaluation of CFUs, TNF- α and LPS in the 2nd efficacy assay using 20 mg/kg of peptide treatment. (**a**) The average CFUs in mice organs (spleen and lung). (**b**) The average concentration of TNF-α in mice serum. (**c**) The average concentration of LPS in mice serum was quantified by ELISA as described in the methodology. All the animals from the colistin group survived until Day 3, so no data for organs and serum analysis can be shown at Day 0. Regarding the vehicle group, no animals survived after Day 0, so no data can be shown at Day 3. Each animal had been inoculated 10^8^ CFU/Kg *A. baumannii* with 5% mucin before treatment. A single treatment was administrated 2 h after the bacteria inoculation. Figure 7b,c were obtained after the correction of OD_450_ with baseline mice, which are neither infected nor peptide treated (only injected at Day 0 with HBS). Mann—Whitney test was used for the comparison between treatments and vehicle except for the additional comparison between ECPep-D and colistin in Figure 7c at day 3 (*** p* < 0.01, ** p* < 0.05, *^#^ p* < 0.1).

**Figure 8 biomedicines-10-00386-f008:**
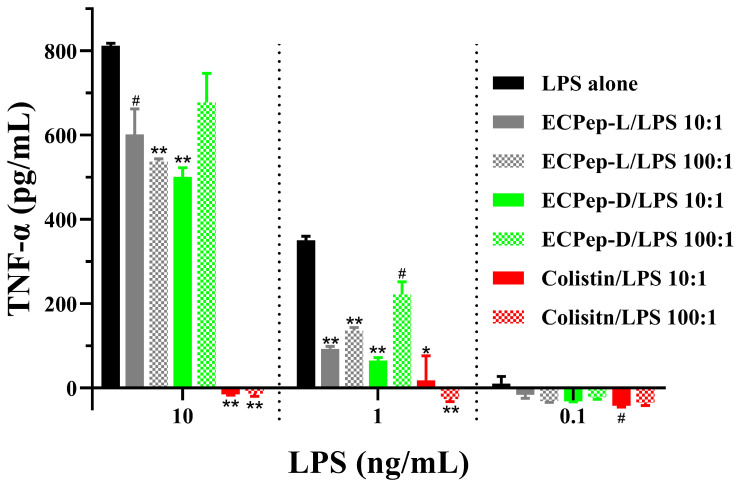
Inhibitory effect on LPS-induced TNF-α cytokine release. Different concentrations of LPS R60-induced secretion of TNF-α by human mononuclear cells and the inhibitory effect in the presence of the peptides ECPep-L, ECPep-D and colistin (positive control) was calculated. Significant differences were estimated in comparison to LPS alone sample (*** p* < 0.01, ** p* < 0.05, *^#^ p* < 0.1).

**Table 1 biomedicines-10-00386-t001:** MIC of N-terminus derivatives of ECP and colistin.

Peptides	MIC ^1^					
	*E. coli*		*P. aeruginosa*		*A. baumannii*	
	μM	μg/mL	μM	μg/mL	μM	μg/mL
ECPep-L	10	37.57	10	37.57	10	37.57
ECPep-D	10	37.57	10	37.57	10	37.57
ECPep-2D-Orn	>20	>70.10	>20	>70.10	5	17.53
Colistin	5	6.34	0.31	0.40	0.31	0.40

^1^ MIC was the concentration with an OD_600_ value equivalent to the OD_600_ value of the bacteria-free control after 24 h incubation. Each MIC was tested in triplicate.

**Table 2 biomedicines-10-00386-t002:** LPS affinity assay of ECP-derived peptides and colistin.

LPS Type			ECPep-L	ECPep-D	ECPep-2D-Orn	Colistin
**EC_50_** ** ^2^ **	**O-LPS**	**μM**	2.71 ± 0.17	2.46 ± 0.19	1.80 ± 0.08	7.87 ± 0.33
		**μg/mL**	10.17 ± 0.62	9.24 ± 0.71	6.30 ± 0.28	9.97 ± 0.42
	**LPSRa**	**μM**	8.87 ± 0.46	7.24 ± 0.70	9.27 ± 0.79	13.12 ± 0.40
		**μg/mL**	33.32 ± 1.72	27.21 ± 2.62	32.50 ± 2.77	16.63 ± 0.50
	**LPSRc**	**μM**	>20	>20	16.56 ± 1.76	19.18 ± 0.34
		**μg/mL**	>75.15	>75.15	58.05 ± 6.16	24.31 ± 0.42

^2^ EC_50_ is the 50% peptide effective concentration, where 50% of the BC probe bound to LPS is displaced. Each group was tested in triplicate and values are presented by Mean ± SE.

**Table 3 biomedicines-10-00386-t003:** Cytotoxic activity of N-terminus derivatives of ECP and colistin.

			ECPep-L	ECPep-D	ECPep-2D-Orn	Colistin
**LD_50_** ** ^3^ ** **(MRC-5)**	**4 h**	**μM**	276.23 ± 25.23	N.D.	N.D.	N.D.
		**μg/mL**	>1000	N.D.	N.D.	N.D.
	**24 h**	**μM**	N.D.	43.51 ± 8.84	>100	>300
		**μg/mL**	N.D.	163.50 ± 33.22	>350.52	>380.26
	**48 h**	**μM**	N.D.	23.52 ± 4.73	>100	>300
		**μg/mL**	N.D.	88.37 ± 17.77	>350.52	>380.26

^3^ LD_50_ is the cytotoxic concentration of the agents to cause death to 50% of viable cells. Each assay was performed in triplicate by the MTT assay and normalized by non-treated control. Values are presented by Mean ± SE. N.D. indicates that no reduction in cell viability was detected at the highest concentration.

## Data Availability

Not applicable.

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
