# Peer review of "In Vivo Evaluation of ECP Peptide Analogues for the Treatment of Acinetobacter baumannii Infection"

_biomedicines, 2022, doi:10.3390/biomedicines10020386_

Round 1
Reviewer 1 Report
Comments
2.12. Quantification of TNF-alpfa LPS by ELISA:The preparation of mononuclear cells should be described.
3.2. In Vitro Activities of ECP Peptides: The results of MIC values (Table 1) and the concentrations of peptides (lines 312-313, Figure 4) are different; the reason should be described.
Figure 8: "Day 0" should be presented as "Day 0 (8 h)", based on the description of the results (page 12, lines 460-471). The effect of colistin should be included in Figure 8a and b at Day 0 (8 h). For the data at Day3, there is no data of vehicle; thus, the effects of peptides could not be evaluated. For Figure 8c, the method for LPS measurement should be included in the methods.
Author Response
We thank the reviewer for the careful revision of our manuscript. We address below all the comments.
1) 2.12. Quantification of TNF-alpfa LPS by ELISA:The preparation of mononuclear cells should be described.
RESPONSE:
The reviewer is right that the text is not clear enough regarding the two assays for quantification of TNF-alpha and LPS. We also include now further details for preparation of mononuclear cells (see sections 2.12 and 2.13: lines 260-274)
2) In Vitro Activities of ECP Peptides: The results of MIC values (Table 1) and the concentrations of peptides (lines 312-313, Figure 4) are different; the reason should be described.
RESPONSE:
The peptide concentrations selected for Figure 4 were chosen in order to better illustrate the differences between the tested peptides. Therefore, we assayed several peptide concentrations below the MIC100 values to visualize the reduction of viability for each bacterial species. To better clarify this point, we have modified the text in order to highlight the differences between Table 1 and Figure 4 conditions (see lines 322-326).
3) Figure 8: "Day 0" should be presented as "Day 0 (8 h)", based on the description of the results (page 12, lines 460-471). The effect of colistin should be included in Figure 8a and b at Day 0 (8 h). For the data at Day3, there is no data of vehicle; thus, the effects of peptides could not be evaluated. For Figure 8c, the method for LPS measurement should be included in the methods.
Figure 8a (the updated number in the new manuscript is Figure 7a) does not include the colistin results at Day 0 because at that time, the percentage of survival was 100% (see Figure 5). Thus, following the ethical procedures, no animals were sacrificed at that time to analyze lung and spleen samples. In the case of the vehicle at Day 3, due the severity of the infection, no animals from the vehicle group were alive at Day 3, so no data can be shown. Regarding the LPS measurement method we have included a more detailed protocol in 2.12 section. Also, in the new manuscript version we have modify Figure 7 legend to include more information.
Reviewer 2 Report
I congratulate authors for this study and for the presented manuscript, which show the efficacy of two N-terminal derived peptides of ECP in a murine systemic A. baumannii acute infection model. The aims and the study design are well-defined. The methods were very appropriate and descriptive. Results were presented in a very clear and organized way. The study has scientific rigor and interest.
Below, I pinpointed just two minor suggestions:
- Line 100: Correct as: American Type Culture Collection (ATCC).
- Lines 137-138: I suggest: “Briefly, bacteria in exponential growth were used to prepare a suspension in MHB with the proximate number of 5 ×105 CFU/mL,…”
Author Response
I congratulate authors for this study and for the presented manuscript, which show the efficacy of two N-terminal derived peptides of ECP in a murine systemic A. baumannii acute infection model. The aims and the study design are well-defined. The methods were very appropriate and descriptive. Results were presented in a very clear and organized way. The study has scientific rigor and interest.
Below, I pinpointed just two minor suggestions:
- Line 100: Correct as: American Type Culture Collection (ATCC).
- Lines 137-138: I suggest: “Briefly, bacteria in exponential growth were used to prepare a suspension in MHB with the proximate number of 5 ×105CFU/mL,…”
RESPONSE:
We thank the reviewer for the positive evaluation of our manuscript. We incorporated the suggestions in the new version of the manuscript.
Reviewer 3 Report
In the following article entitled “In Vivo Evaluation of ECP Peptide Analogues for the Treatment of Acinetobacter baumannii Infection”, the authors describe a complete analysis of two novel eosinophil cationic protein (ECP) in comparison to colistin. It is an important research manuscript, with a broad and detailed introduction. Importantly, it is a complete story from chemistry to in vivo results.
I attached the manuscript to highlight as many points as possible.
Likely, one missing point is the interpretation in regards to O-LPS binding that is similar between the L- and D-peptides. Any future directions might be interesting to add in the discussion since it is rather unusual.
Overall, the early part of the manuscript always refers to ECPep-L (MIC, Binding, Toxicity...); then this link with the original peptide is lost. If available, it will be nice to include in the text to help with the data analysis and interpretation (also for reader to measure the impact of the retro peptide).
All the figures and tables are clear; albeit, it would be great to put the figure 5 in supplementary information. I would also appreciate the AA numbers on figure 2, the 3 CD traces for figure 3 and the peptide characterizations (RP-HPLC and MS) in supplementary information.

Author Response
In the following article entitled “In Vivo Evaluation of ECP Peptide Analogues for the Treatment of Acinetobacter baumannii Infection”, the authors describe a complete analysis of two novel eosinophil cationic protein (ECP) in comparison to colistin. It is an important research manuscript, with a broad and detailed introduction. Importantly, it is a complete story from chemistry to in vivo results.
I attached the manuscript to highlight as many points as possible.
We thank the reviewer for the careful revision of our work. We indicate below the point by point response.
1) Likely, one missing point is the interpretation in regards to O-LPS binding that is similar between the L- and D-peptides. Any future directions might be interesting to add in the discussion since it is rather unusual.
We have added some discussion on that issue (see lines 348-351 and 670-672).
2) Overall, the early part of the manuscript always refers to ECPep-L (MIC, Binding, Toxicity...); then this link with the original peptide is lost. If available, it will be nice to include in the text to help with the data analysis and interpretation (also for reader to measure the impact of the retro peptide).
The half-life in serum of the ECPep-L is very short (see Sandín et al. JMC 2021), so it is not suitable for in vivo studies. On the other hand, the physico-chemical properties and antimicrobial activity of ECPep-L in vitro were previously well characterized. We have revised the manuscript to properly refer now to all the work done for each of the peptides and to justify the selection of the modified analogues for the in vivo assays. We added now a paragraph at the end of section 3.2.
3) All the figures and tables are clear; albeit, it would be great to put the figure 5 in supplementary information. I would also appreciate the AA numbers on figure 2, the 3 CD traces for figure 3 and the peptide characterizations (RP-HPLC and MS) in supplementary information.
As suggested, we have transferred Figure 5 to the supplementary information (now figure S7).
We added the amino acid number of each peptide in figure 2.
CD traces for ECP-L and ECPep-2D-Orn have been compared in the previous study (Sandín et al. 2021, Supplementary Figure 6A and C). We include now the information within the text.
We also added the RP-HPLC and MS chromatograms in supplementary materials (see new Figure S2).
Round 2
Reviewer 1 Report
Comments
- For the preparation of mononuclear cells, the reference should be included.
- For the measurement of LPS, the detection mechanism of LPS and the kit of LPS ELISA should be described. The reviewer is wondering whether the LPS in sera can be measured by ELISA, because the serum components may affect the reactions for LPS detection. The authors stimulated mononuclear cells by 10 ng/ml LPS. However, in mouse sera, microgram levels of LPS were detected; the LPS concentrations are too high in sera.
Author Response
See attached pdf file.

Round 3
Reviewer 1 Report
The authors have almost properly responded to the comment sof the reviewer.